# TEST-TIME RAG (TTRAG): ENHANCING LONG CONTEXT UNDERSTANDING IN LLMS WITH RETRIEVAL-AUGMENTED MECHANISMS

## ABSTRACT

Large Language Models (LLMs) are becoming increasingly pivotal in applications that depend on extensive personalized context, such as conversational agents and specialized task-oriented systems. In these scenarios, effective long-context handling is essential to support agentic tasks and enhance in-context learning capabilities. To address this challenge, we propose a novel integration of Retrieval-Augmented Generation (RAG) techniques with LLMs, designed to enhance their ability to effectively manage and utilize large contextual information only available at test time. Our methodology, Test-Time RAG (TTRAG), enriches LLMs by dynamically generating novel conditional embeddings coupled with query rewriting and utilizing semantic search to retrieve the most relevant document chunks at test time. This process preserves the context's meaning and enhances the model's responsiveness and accuracy in knowledge-intensive Question Answering (QA) tasks. Our evaluations demonstrate our system's ability synthesize and retrieve information across extensive texts: HotpotQA (+17.29%), QASPER (+4.39%), and Natural Questions (+8.73%), demonstrating the effectiveness of TTRAG across varied context lengths from 1 million to 9.6 million tokens.

## 1 INTRODUCTION

The ability for LLMs to incorporate a detailed history of a user's experiences and actions could unlock a new era of personalized intelligence. For example, an LLM helping a user respond to an email could take into account much more than just the text in the email: it could also take into account the user's documents, calendar schedule, and other related emails or shared data. However, capturing this scale of rich context for a user presents several challenges. First, because the information is diverse and sparse, naively retrieving related documents by keyword or vector similarity alone is insufficient (Karpukhin et al., 2020; Khattab & Zaharia, 2020). Second, the potentially relevant context could be orders of magnitude larger than what can fit within existing long-context models (Anthropic, 2023; Reid et al., 2024). Finally, when the information does fit in context, the use of long-context and the presence of distracting information can lead to significant quality loss and increased latency (Liu et al., 2023c).

Retrieval Augmented Generation (RAG) systems have been increasingly employed for enabling LLMs to effectively access large collections of data (Lewis et al., 2021; Borgeaud et al., 2022; Wang et al., 2023; Patil et al., 2023). RAG operates by first computing document embeddings for each document in the collection and then computes a similarity score with a given prompt to determine a set of documents that should be included as context in a prompt to an LLM. RAG systems must pre-compute document embeddings, limiting their application to scenarios where documents cannot be indexed in advance. Such situations are common in our personalized applications including chat-based interactions (Peng et al., 2023), code analysis, database joins, and privacy-centric systems where personal data is not available offline. To overcome these constraints and unlock new possibilities in long-context processing, this paper presents a systems that extends classic RAG techniques of embedding and filtering to a *test-time* setting, which we refer to as *Test-time RAG* (TTRAG).

As illustrated in fig. 1, TTRAG is built upon three critical components that address the challenges posed by dynamic knowledge bases and enhance the efficiency of long-context processing: *Con-*

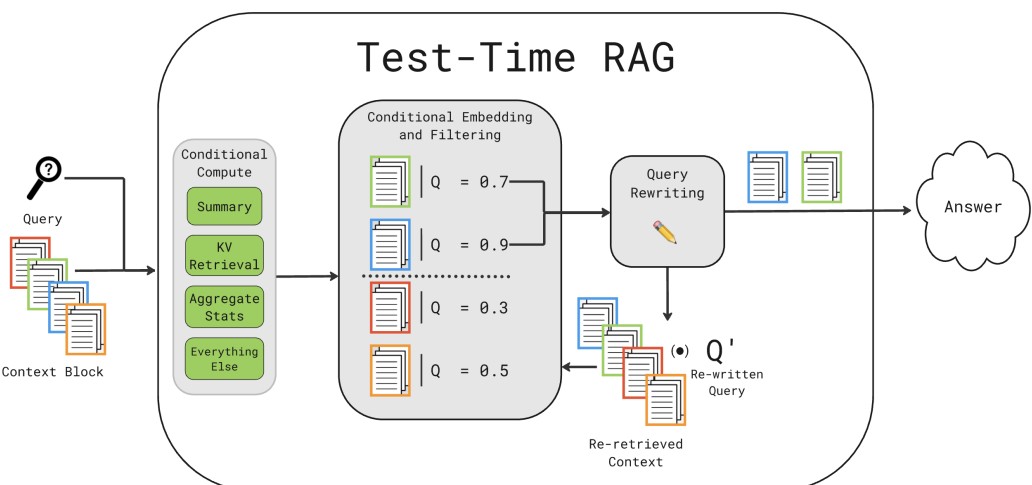

Figure 1: Test-time RAG (TTRAG) operates by first employing conditional compute to determine the appropriate set of computations to perform on the given query (see §3.3 for details.) For most queries, TTRAG proceeds with conditionally embedding the query to an embedding space, and then using the embedded values to filter a set of relevant documents. TTRAG then iteratively rewrites the query and re-retrieves documents, generating a response utilizing relevant documents once the golden context is found.

*ditional embeddings*, a technique for dynamically generating document representations that are conditioned on the specific user query, which leads to more contextually appropriate embeddings. *Iterative query rewriting*, to better capture user intent and refine the search process, our system employs an iterative query rewriting mechanism that is conditioned on both, the user prompt, and documents retrieved so far. This dynamic refinement enables the system to expand or adjust user queries as needed, improving the quality of document retrieval and enhancing the overall understanding of the user's request. Finally, inspired by classical distributed systems literature (Dean & Ghemawat, 2008), our TTRAG system employs *conditional compute* to adapt the computation, adjusting the processing depth and complexity based on the specific requirements of each query. This targeted approach optimizes resource use without compromising output quality.

We evaluate the TTRAG pipeline on a variety of knowledge-intensive, question-answer datasets including HotPotQA (Yang et al., 2018), Natural Questions (Kwiatkowski et al., 2019), Comprehensive RAG benchmark (CRAG) (Yang et al., 2024), QASPER (Dasigi et al., 2021), QuALITY (Pang et al., 2022). We additionally source tasks from InfiniteBench (Zhang et al., 2024) to evaluate on a diverse set of long context tasks outside the scope of question answering. These datasets span multiple domains and tasks that require reasoning over long context, providing a comprehensive assessment of our system's capabilities. TTRAG is benchmarked against state-of-the-art long context models to demonstrate the performance and quality improvements achieved by our proposed approach over current paradigms of long-context processing, and demonstrate significant improvements in LLM performance in all our evaluations.

By enabling AI systems to handle dynamic and changing knowledge bases, Test-time RAG opens up new possibilities for embedding personalized intelligence in user-applications over rapidly evolving contexts. Furthermore, Test-time RAG does not require pre-indexing of sensitive information until and unless it is necessary, making it particularly suitable for privacy-aware applications in healthcare, finance, and other domains where users might not prefer to have their personal data indexed and stored. To summarize, this paper makes the following contributions:

1. Propose three test-time compute techniques for RAG: conditional embeddings, iterative query rewriting, and conditional compute

2. Introduce an end-to-end system which ensembles these test-time compute techniques and scales performant long-context processing to millions of tokens

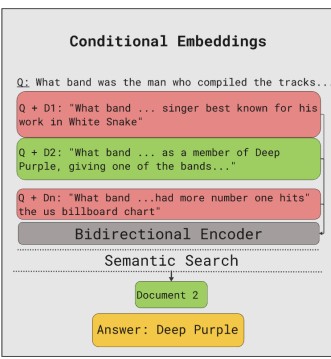 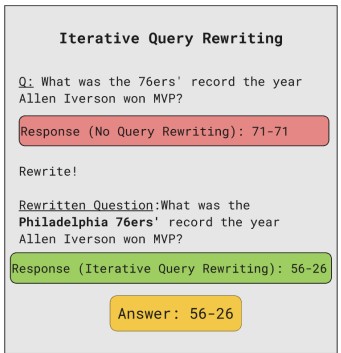 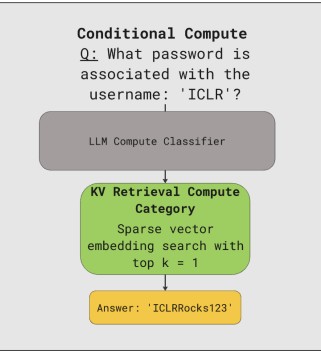

Figure 2: **TTRAG encompasses test-time embedding, rewriting, and task-conditional compute, handling the entire breadth of long context tasks and questions.** Because these techniques are applied at inference time, there is no additional overhead to integrating these techniques with existing popular language models. Boxes shaded in green correspond to correct responses from TTRAG, while red boxes correspond to incorrect responses or selected documents from our baseline formulations. Golden boxes represent the correct ground truth answer.

> 3. Demonstrate that Test-time RAG improves knowledge intensive QA by upto +17.29% on GPT-3.5-turbo for HotpotQA

## 2 MOTIVATION

While in-context learning has proved to be a powerful tool for providing the model with relevant context (Wies et al., 2024), LLMs that support long-contexts face three challenges: the computational overhead required for training, the degradation in performance at longer-context, and the latency overhead at inference time. The self-attention mechanism in auto-regressive transformer-based architectures suffers from quadratic increase in complexity as context length increases (Vaswani, 2017). This computational burden poses a substantial challenges in training larger context models Dubey et al. (2024). Moreover, even when long context capabilities are implemented, models often experience performance degradation (Li et al., 2024; Liu et al., 2023c), as well as increased latency (Agrawal et al., 2024), as the context window expands, limiting their practical effectiveness.

Classically, to address these limitations, two techniques are adopted: retrieval augmented generation (RAG) and recursive summarization. These approaches aim to filter content, and compress information to fit within the limited context windows of current models respectively. Recursive summarization (Ren et al., 2017), while effective in some scenarios, inherently results in lossy compression of previous context, often sacrificing important information as the context grows. RAG (Lewis et al., 2020), on the other hand, pre-indexes a domain of context into an external database of documents and employs information retrieval mechanisms to fetch relevant context at prompt time. While RAG has shown promise in many applications, its effectiveness is heavily dependent on having prior knowledge of the information set required to answer prospective queries. None of the solutions mentioned before provide an effective technique to handle large amounts of data introduced at test time and do not exploit the availability of intent present in the prompt.

Other efforts have looked at advancements in long context processing driven by architectural modifications such as State-space models (Gu et al., 2021), ring attention (Liu et al., 2023a), sparse attention (Chen et al., 2021; Zoph, 2022), etc. These advancements are complementary and benefit from Test-time RAG.

## 3 TEST-TIME RAG

Test-time RAG (TTRAG) contains three primary components that work in conjunction: conditional embeddings, iterative query rewriting, and conditional compute. TTRAG is an ensemble, with each component contributing individual merits that fit together as a compound, yet singular system.

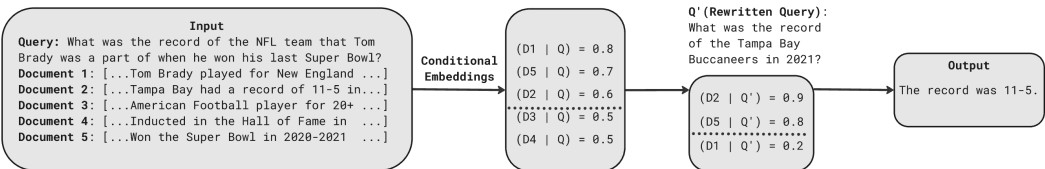

Figure 3: End-to-end example demonstrating TTRAG. The scores shown following the conditional embeddings represent similarity scores between the embedding and query.

### 3.1 CONDITIONAL EMBEDDINGS

We present a novel approach to generate conditional embeddings at test time that capture detailed semantic relationships between context and query. The fundamental mechanics which power conditional embeddings rely on the premise enforced in test-time settings: access to the user query. Traditional approaches to compute document embeddings are done offline and independent of the queries themselves. We instead propose a method to compute document embeddings on a per-query basis which leads to selecting documents that are more related to the query during retrieval. Our approach contains two components that utilize query information: conditional embedding generation and filtering. During the embedding generation step, we employ the query to tailor granular representations of text such that the filtering portion can efficiently retrieve relevant context over large contexts. To accomplish this, we leverage the contextual embedding capabilities that are implicitly present in the transformer architecture, and utilize encoder-decoder variants which jointly condition context on text appearing before and after a given token in a sequence (Devlin et al., 2019).

Decoder-only architectures where masked, causal attention is utilized, only allow tokens to attend to previous tokens in the sequence, with next token prediction as the training objective (Radford et al., 2019). In contrast, bidirectional encoders allow each token to attend to all other tokens, creating token representations that have contextual awareness of other tokens in the sequence regardless of their ordering as they pass through the encoder (Devlin et al., 2019). Bidirectional attention is a critical component of conditional embeddings generation to enable proper conditioning on the query. Conditional document embeddings are constructed by first prefixing each document with the query text. This augmented document is passed through a bidirectional encoder (DistilBERT), where we extract the final hidden state embeddings, mask the tokens corresponding to the prefixed query (visualized in fig. 4), then take an average across the sequence length dimension of the unmasked token embeddings. From our ablations, we find that the performance of embeddings generated from pooling the last hidden state is agnostic of the presence of the <EOS> token, which we leave unmasked. We are left with a fixed length embedding that captures fine grained relationships between the query and context, which can be used in similarity search without the prefixed query inflating the similarity scores. Conditional Embeddings (Filtering only) applies cosine similarity search between query and document embeddings to only retain semantically similar context documents.

### 3.2 ITERATIVE QUERY REWRITING

Drawing inspiration from query reformulation techniques in recommendation systems (Bhandari et al., 2023), we introduce a framework for automatic query rewriting to improve retrieval and generation quality. Questions over long context often have vague subjects and directives, as these aspects are meant to be derived from context. However, in retrieval augmented paradigms, this poses as a bottleneck during filtering and similarity search, as the standard embeddings struggle with capturing the fine-grained semantics of a vague query with respect to a set of documents. We leverage reasoning capabilties of LLMs to make the query more conducive for similarity search. We first pass the query along with the top *three* (hyperparameter) documents retrieved from similarity search to the LLM. Rather than immediately moving to the answer generation step, we prompt the LLM to determine whether the answer to the query is found in the provided context. If the answer is not found, the LLM is tasked with rewriting the query to be a better proxy for retrieving relevant context given the documents provided. During this step, the LLM has freedom to incorporate additional information from the documents into the rewritten question to dispel ambiguity and reword the question structure so the question directive is less vague. If the answer is found, the LLM is instructed to just return the

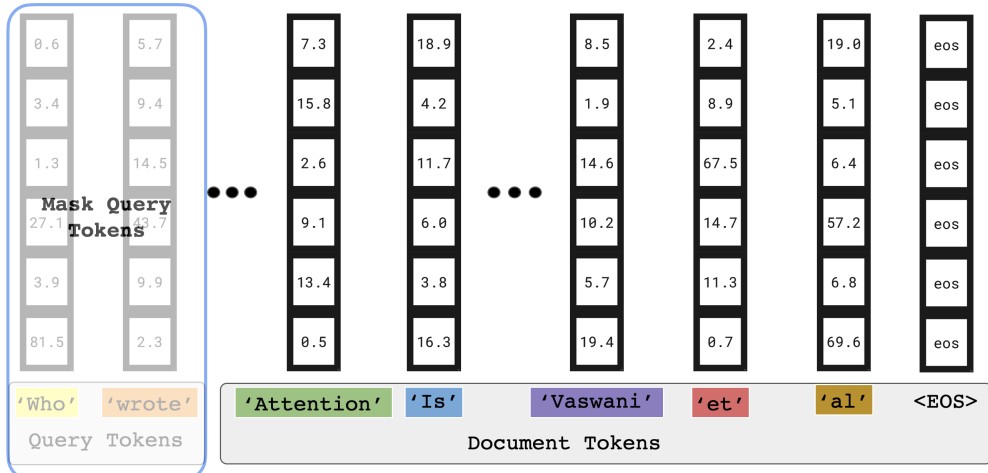

Figure 4: Masking the query tokens prior to mean pooling allows document token representations to retain conditional query information without inflating similarity scores during similarity search.

answer. The query rewriting step is repeated until we have retrieved a total of k (hyperparameter) documents across all iterations.

---

**Algorithm 1** Iterative Query Rewriting Pseudocode

---

1:  **while** number of iterations < maximum rewrite tries **do**
2:      Compute query and document embeddings
3:      retrieved docs = top n documents from similarity search (parameter)
4:      Prompt LLM to determine if answer in context
5:      **if** answer in context **then**
6:          **return**  answer
7:      **end if**
8:      **if** answer not in context **then**
9:          LLM rewrites query
10:     **end if**
11: **end while**
12: **return**  retrieved docs

---

TTRAG's online query-rewriting surfaces the following benefits: First, an observed improvement in the LLM's ability to parse context, which would be otherwise considered complex, highlighted in fig. 9 and fig. 10. Second, we demonstrate in fig. 8 scenarios where the LLM blends parametric knowledge from pretraining with the non-parametric knowledge from provided context more effectively which would otherwise not have been possible. Lastly, when used as a component in Test-time RAG, all of the aforementioned benefits hold, while also decreasing the total number of tokens processed by the LLM per query.

## 3.3 CONDITIONAL COMPUTE

Access to the query and documents at test time allows our pipeline to incorporate a conditional compute feature designed to minimize compute while maintaining output quality based on the nature of the user's query, which greatly enhances its versatility across various tasks such as summarization, question answering and key-value retrieval. Categorizing different user questions into various "compute categories" is built on the idea that different types of information require distinct strategies for document retrieval and processing. Adding this level of granularity results in context retrieval that is more cost-effective and compute-efficient. In this paper, conditional compute refers to the number of tokens processed at test time conditioned on the query, consistent with literature (Brown, 2021)

**Task Classification and Response Generation:** Each user query is classified into four predefined categories: summarization tasks, KV retrieval tasks, aggregate statistics queries, and question-answering tasks by an LLM classifier. Depending on the determined category, the pipeline adjusts its subsequent processing steps to tailor the retrieval and response generation specifically to the task at hand.

**Summary Tasks:** Retrieval based approaches historically struggle with creating summaries due to the lack of whole context comprehension. For queries categorized under summarization, we split the large context block into smaller batches that fit into the context window of the base LLM and then ask the LLM to generate summaries for these smaller chunks and concentenate them to form the output.

**Aggregate Statistics Tasks:** We adopt a selective filtering-based implementation that identifies the target block of text and calculate relevant aggregate statistics across the entire document set.

**KV Retrieval Tasks:** Key-value pair (KV) retrieval challenges the LLM with accurately retrieving a value given a key from a large block of JSON objects (Liu et al., 2023c). The complexity of this task is lies in the large retrieval base and the indistinguishable format of relevant and irrelevant information, making the actual textual content the only distinct attribute. To tackle these challenges, we employ a sparse BM25 retriever to fetch the most similar document from the context (top k = 1).

**Question-Answering Tasks:** For queries that do not fall into any of the other categories, the pipeline defaults to our standard Test-time RAG retrieval process, leveraging the configured embedding model to fetch documents that best match the query's semantic content.

## 4 EXPERIMENTS AND RESULTS

The efficacy of TTRAG is assessed by observing performance of the system as a whole and individually testing core components on a wide variety of datasets, utilizing open and closed-source models. We seek to answer the following questions:

1. What are the performance improvements gained from TTRAG as a complete system?
2. What are the quantified contributions from each component?
3. What is the reduction in compute at test-time?
4. How generalizable are these performance gains across different datasets and domains?

### 4.1 DATASETS, MODELS

We center our findings around the end-to-end evaluation of TTRAG and bolster them by presenting ablation studies to quantify individual component performance, using a common base of models and datasets. All datasets have long documents as context, either by nature of the dataset or by artificially augmenting context during preprocessing.

**Conditional Embeddings** Conditional embedding performance is evaluated on two different categories of datasets: general question-answering spanning multiple domains and domain-specific datasets. For the first group of datasets, we use popular knowledge intensive question-answer datasets: HotpotQA, Natural Questions, and PubMedQA (Yang et al., 2018; Kwiatkowski et al., 2019; Jin et al., 2019).

For HotpotQA and Natural Questions, we extend the model-provided context to millions of tokens by artificially concatenating documents across all questions in their respective datasets. We additionally evaluate conditional embeddings on a second group of datasets with domain-specific context to test if conditional embeddings are proficient at creating granular representations of text from context blocks that are all relatively similar to each other. QASPER is a dataset consisting of question and answering over NLP research papers, while QuALITY requires comprehension over whole text passages Dasigi et al. (2021); Pang et al. (2022). Both datasets contain approximately 5k tokens of context on average per question, all pertaining to the same body of text.

**Iterative Query Rewriting** CRAG, a comprehensive RAG-specific benchmark, presents a unique challenge for query rewriting by including ~300k tokens of simulated search results per question (Yang

et al., 2024). In comparison to the experimental evaluations done with HotpotQA, the information contained in CRAG question documents is related. The inherent similarity between documents can therefore be used as a proxy to determine if rewritten queries result in more nuanced retrieval.

**Conditional Compute** The InfiniteBench suite of evaluations contains a wide breadth of long context tasks outside of traditional question answering, and is used to evaluate conditional compute. En.Sum is used to benchmark summarization capabilities and Retrieve.KV tests the synthetic KV retrieval task (Zhang et al., 2024).

**Models and Baselines** We evaluate our augmented LLM approach using various models (reported along with context): GPT-4-0613 (8k), GPT-3.5-turbo-0125 (16k), Claude Haiku (200k), Llama2-7B Chat (4k), and GPT-4-turbo-2024-04-09 (128k)

We also evaluate performance with two different styles of retrieval: BM25 – sparse vector representation (Robertson & Zaragoza, 2009), and OpenAI's text-embedding-3-small – LLM based dense vector representation. For our baselines, we use the same retriever as our pipeline to select the top 10 most relevant documents and backfill the rest of the context window with random documents from the dataset, acting as distractor documents. Moreover, the documents in the context window are randomly shuffled. This formulation ensures that golden documents are present in our baselines while saturating the context window, which evaluates how well the baseline is able to process information at maximum token input capacity. We additionally include performance of conditional embeddings (Filtering only) configured with either dense or sparse vector embedding models in our ablations to further establish performance of TTRAG's components over standard retrieval strategies.

## 4.2 TEST-TIME RAG

Table 1: **TTRAG improves exact matching accuracy by 17.29% when compared to baselines, and demonstrates significant gains across diverse datasets.** Each component of TTRAG stacks performance benefits in both open and closed source settings. CE refers to conditional embeddings isolated as an individual component.

| Model | | HQA(1M) | | QASPER | | QuALITY | | NQA(5M) | |
|---|---|---|---|---|---|---|---|---|---|
| Name | Embedding Model | Exact Match | F1 Score | Exact Match | F1 Score | Exact Match | F1 Score | Exact Match | F1 Score |
| GPT-3.5-0125 Baseline | text-embedding-3-small | 41.18% | 36.04 | 22.75% | 29.0 | 53.48% | 51.20 | 36.62% | 38.29 |
| GPT-3.5-0125 CE | text-embedding-3-small | 51.76% | 47.46 | 25.90% | 30.33 | 59.57% | 55.97 | 42.69% | 36.43 |
| **GPT-3.5-0125 TTRAG** | **text-embedding-3-small** | **58.47%** | **48.83** | **27.14%** | **29.7** | **64.35%** | **59.18** | **45.35%** | **33.93** |
| Llama-2-7b-chat-hf-Baseline | text-embedding-3-small | 27.18% | 11.20 | 16.04% | 9.60 | 14.78% | 2.71 | 22.58% | 13.75 |
| Llama-2-7b-chat-hf CE | text-embedding-3-small | 38.24% | 17.83 | 18.92% | 10.91 | 16.09% | 2.24 | 26.76% | 17.01 |
| **Llama-2-7b-chat-hf TTRAG** | **text-embedding-3-small** | **44.35%** | **30.04** | **19.72%** | **19.26** | **23.91%** | **4.62** | **30.55%** | **22.2** |

As a whole, TTRAG outperforms baselines in a variety of datasets when applied to closed and open-sourced models, shown in table 1. We include a comparison of the results with conditional embeddings as an isolated component of the TTRAG system to highlight that the collective arrangement of all components in TTRAG leads to superior performance. The configuration of TTRAG for question-answering tasks employs conditional embeddings to power iterative query rewriting. Specifically, during each iteration of rewriting the query, conditional embedding and filtering is used to retrieve the top k most similar documents. We fix the conditional compute component to question-answering mode in order to match the nature of datasets we evaluate on.

## 4.3 CONDITIONAL EMBEDDING

We evaluate TTRAG's first component, conditional embeddings, over datasets that are state-of-the-art in the respective domains of domain-specific question-answering tasks (QASPER, QuALITY) and general knowledge question-answering tasks (HotPotQA, Natural Questions).

**Domain Specific Question-Answering Tasks** From fig. 7, we demonstrate that conditional embeddings serve as a powerful re-ranker boosting the model's performance from 22.75% to 25.90% for GPT-3.5-turbo-0125 on QASPER table 2 and from 53.48% to 59.57% on QuALITY in table 6 (moved to Appendix for space). We additionally note three further observations: first, conditional embeddings demonstrate improved performance when used with bm25 and text-embedding-3-small.

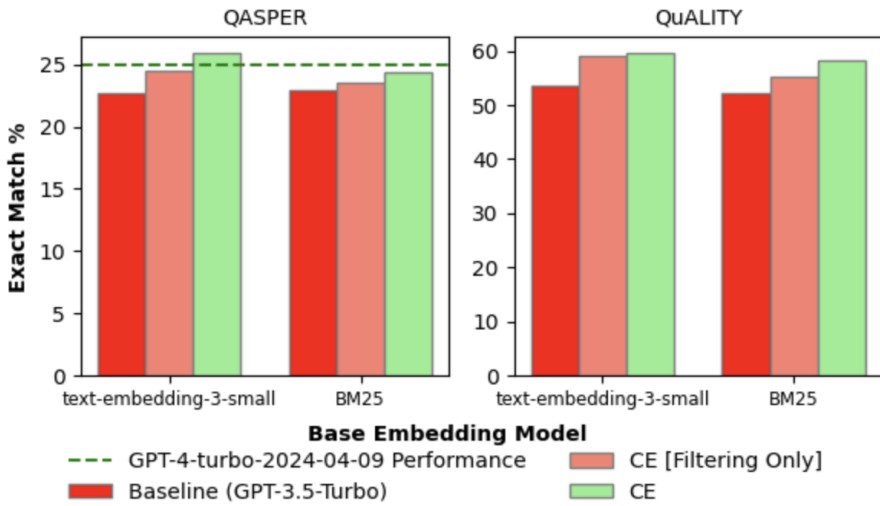

Figure 5: **Conditional embeddings improves accuracy over baseline with dense and sparse embedding models.** The performance of GPT-3.5-turbo coupled with conditional embeddings surpasses performance of a much stronger GPT-4-turbo on QASPER by 0.9% and achieves a 3.5% lift over its baseline.

Second, the performance contributions from conditional embeddings alone boosts performance of a substantially weaker base model GPT-3.5-turbo-0125 to match performance of much stronger models, GPT-4-turbo-2024-04-09 on QASPER, shown in fig. 5. Finally, we demonstrate that the trends hold even when ported to Llama-models "as-is".

Table 2: **Conditional embeddings performance is agnostic of base model or embedding model.** Between all model permutations run on QASPER, conditional embeddings results in a 3% improvement from baseline on average.

| Model | | QASPER | |
| --- | --- | --- | --- |
| Name | Embedding Model | Exact Match | F1 Score |
| GPT-4-turbo-2024-04-09 Baseline | text-embedding-3-small | 25.00% | 25.58 |
| GPT-3.5-0125 Baseline | text-embedding-3-small | 22.75% | 29.0 |
| GPT-3.5-0125 CE (Filtering only) | text-embedding-3-small | 24.44% | 29.89 |
| **GPT-3.5-0125 CE** | **text-embedding-3-small** | **25.90%** | **30.33** |
| GPT-3.5-0125 Baseline | bm25 | 22.97% | 27.99 |
| GPT-3.5-0125 CE (Filtering only) | bm25 | 23.54% | 26.67 |
| **GPT-3.5-0125 CE** | **bm25** | **24.32%** | **29.23** |
| Llama-2-7b Baseline | text-embedding-3-small | 16.04% | 9.60 |
| Llama-2-7b CE (Filtering only) | text-embedding-3-small | 16.22% | 10.1 |
| **Llama-2-7b CE** | **text-embedding-3-small** | **18.92%** | **10.91** |
| Llama-2-7b Baseline | bm25 | 13.74% | 9.87 |
| Llama-2-7b CE (Filtering only) | bm25 | 15.54% | 10.11 |
| **Llama-2-7b CE** | **bm25** | **18.81%** | **10.24** |

**General Knowledge Question-Answering Tasks**   On general knowledge question-answering tasks HotPotQA, Natural Questions, and PubMed ( table 7, and  table 8 of Appendix respectively),  table 3 highlights the importance of aggressive filtering. We demonstrate that in Test-time RAG, aggressively dropping documents even when additional context length is permissible is recommended. While it may seem counter-intuitive to possibly give up recall (potentially impacting precision), distractor documents actually hurt the model in comprehension tasks even when the model is trained for longer context lengths. Test-time RAG's consistent performance over HotPotQA  table 3, Natural Questions  table 7, and PubMed  table 8 highlight that with the *conditional embeddings* component of Test-time

RAG alone, classical short-context length models can be used at upto 8M+ context lengths without significant loss in performance.

Table 3: Conditional Embeddings outperforms baselines at million token context lengths for knowledge intensive question-answering tasks. With Test-time RAG's, we are able to scale LLMs at context lengths up to 8.8M million tokens even with BM25 while retaining performance on HotPot QA (HQA). Natural Questions and PubMed included in table 7, table 8 of Appendix.

| Model | | HQA(1M) | | HQA(5M) | | HQA(8.8M) | |
|---|---|---|---|---|---|---|---|
| Name | Embedding Model | Exact Match | F1 Score | Exact Match | F1 Score | Exact Match | F1 Score |
| GPT-4-0613 Baseline | text-embedding-3-small | 66.12% | 34.25 | 64.29% | 32.58 | 61.24% | 31.66 |
| GPT-4-0613 CE (Filtering only) | text-embedding-3-small | 69.41% | 37.52 | 65.07% | 34.29 | 62.22% | 32.98 |
| **GPT-4-0613 CE (Filtering only)** | **bm25** | **70.12%** | **38.19** | **67.36%** | **35.87** | **67.25%** | **36.22** |
| GPT-4-turbo-2024-04-09 Baseline | text-embedding-3-small | 70.00% | 48.14 | 63.88% | 45.62 | 60.63% | 38.96 |
| GPT-4-turbo-2024-04-09 CE (Filtering only) | text-embedding-3-small | 73.06% | 51.01 | 64.63% | 47.30 | 63.00% | 45.01 |
| **GPT-4-turbo-2024-04-09 CE (Filtering only)** | **bm25** | **73.65%** | **53.79** | **70.38%** | **51.43** | **70.5%** | **49.9** |
| Llama-2-7b-chat-hf-Baseline | text-embedding-3-small | 27.18% | 11.20 | 26.56% | 10.51 | 26.94% | 11.41 |
| Llama-2-7b-chat-hf CE (Filtering only) | text-embedding-3-small | 35.76% | 12.46 | 32.56% | **13.48** | 31.06% | **14.43** |
| Llama-2-7b-chat-hf CE (Filtering only) | bm25 | **36.35%** | **12.65** | **33.65%** | 12.73 | **32.00%** | 12.72 |
| Claude-3-Haiku-Baseline | text-embedding-3-small | 57.82% | 21.44 | 55.56% | 21.28 | 48.67% | 21.65 |
| Claude-3-Haiku CE (Filtering only) | text-embedding-3-small | **54.35%** | 42.57 | 51.04% | 38.37 | 50.1% | 37.91 |
| Claude-3-Haiku CE (Filtering only) | bm25 | 53.18% | **44.53** | 52.59% | 43.68 | 52.73% | 44.27 |

## 4.4 QUERY REWRITING

Evaluations on query rewriting suggest material gains in accuracy, demonstrated in table 4. More comprehensive context parsing and inclusion of an LLM's parametric memory is observed in examples fig. 8, fig. 9, fig. 10. From a compute efficiency standpoint, **the number of tokens processed by the LLM is less when compared to just conditional embeddings (Filtering only) while improving performance**, due relevant context being found faster during the iterative query rewriting algorithm, shown in fig. 6. Additionally, the number of documents shown to the LLM is upper bounded by the top k set in conditional embeddings (Filtering only), establishing query rewriting as more token compute friendly. As shown in table 5, the average number of query rewrites is minimal for correctly answered questions in CRAG, supporting that the inclusion of less than 2 iterations of rewriting results in less tokens processed when compared to conditional embeddings (Filtering only). To isolate the query rewriting component of TTRAG from conditional embedding generation, we utilize conditional embeddings (Filtering only) to retrieve out-of-the-box embeddings generated from text-embedding-3-small during each iteration of query rewriting. We then benchmark against conditional embeddings (Filtering only) without query rewriting alongside our standard baseline formulation.

Table 4: **The performance gains in exact match accuracy from query rewriting are present in both large and small models**. GPT-3.5-turbo has approximately 175 billion parameters while Llama2 and Llama3 have 7 and 8 billion parameters respectively. Therefore even small models with significantly less reasoning capabilities (which directly correlates to the quality of rewriting) concretely benefit from query rewriting.

| Model | | CRAG | |
|---|---|---|---|
| Name | Embedding Model | Exact Match | F1 Score |
| **GPT-3.5-0125 Query Rewrite** | **text-embedding-3-small** | **30.04%** | **23.53** |
| GPT-3.5-0125 CE (Filtering only) | text-embedding-3-small | 28.94% | 23.69 |
| **Llama-2-7b Query Rewrite** | **text-embedding-3-small** | **21.18%** | **21.83** |
| Llama-2-7b CE (Filtering only) | text-embedding-3-small | 20.33% | 21.26 |
| **Llama-3-8B Query Rewrite** | **text-embedding-3-small** | **23.47%** | **20.77** |
| Llama-3-8B CE (Filtering only) | text-embedding-3-small | 23.02% | 20.25 |

## 4.5 CONDITIONAL COMPUTE

Evaluation for conditional compute is carried out independently per compute category. For key-value retrieval task, when presented with an effective context window of 20 million tokens, our pipeline

Table 5: **The number of documents processed to generate an answer with query rewriting is strictly less than naively retrieving 5 documents via conditional embeddings (Filtering only).** In our evaluations each iteration of rewriting processes 3 documents.

| Model | Average Number of Rewrites |
|---|---|
| GPT-3.5-0125 | **1.22** |
| Llama-2-7B | 1.42 |
| Llama-3-8B | 1.31 |

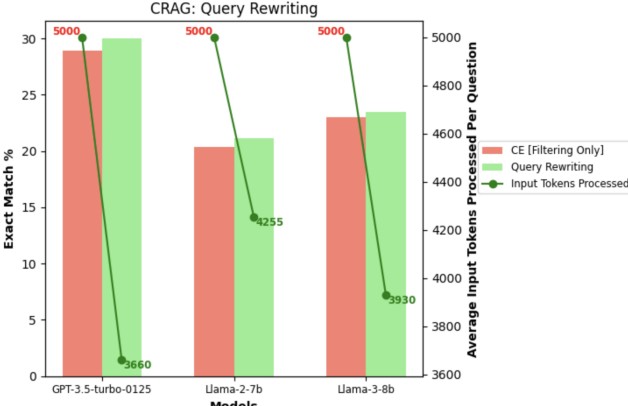

Figure 6: **Query rewriting improves accuracy across closed and open sourced models and finds golden documents faster.** Due to the iterative structure of query rewriting, the answer converges with a 20% reduction of tokens processed compared to conditional embeddings (Filtering only) and with higher accuracy.

has an exact match score of 85.2%, which far exceeds the capabilities of available currently LLMs at this context size. Table 10 enumerates our latency and compute on the KV retrieval task. For summarization, conditional compute achieves comparable results (32.21 F1 score and 8.65 ROUGE-L score) while only processing 25k total tokens as opposed to the 171k tokens of context provided. Table 11 and fig. 11 outline compute efficiency gains for summarization. When viewing aggregate results displayed in fig. 12 , conditional compute processes 10x less tokens on average, which is a drastic efficiency gain in long-context settings.

## 5 CONCLUSION

In conclusion, this paper presents Test-time RAG, a novel system incorporating conditional embeddings, iterative query rewriting, and conditional compute. As a novel extension to classical RAG, TTRAG addresses the challenges of incorporating vast, dynamic knowledge bases in personalized AI applications. Test-time RAG enables more flexible and efficient use of long contexts, overcoming the limitations of pre-indexing and static retrieval mechanisms. Our evaluation of TTRAG is two-fold: we measure the performance of the system as a whole, then present ablations exploring the individual contributions of each component across various knowledge-intensive, domain-specific, and long-context benchmarks, including HotpotQA, Natural Questions, PubMed, QASPER, QuALITY, and CRAG. Test-time RAG demonstrates consistent and substantial improvements in accuracy and context-handling capabilities with varied base models in diverse settings, showcasing TTRAG's potential in scaling long-context processing to millions of tokens. By dynamically adapting at test time, Test-time RAG empowers large language models to personalize and refine their responses based on evolving user contexts, enhancing their utility in real-world, user-focused scenarios.

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

# A APPENDIX

## A.1 CONDITIONAL EMBEDDING EXAMPLE

We include an example in fig. 7, demonstrating a scenario where conditional embeddings excels over traditional embedding methods.

```
Question: What band was the man who compiled the tracks for The Early
Years a part of when they were inducted into the Rock and Roll Hall of
Fame?

Without Conditional Embeddings:
Response: White Snake

With Conditional Embeddings:
Retrieved Documents: David Coverdale (born 22 September 1951) is an
English rock singer best known for his work with ... In 2016,
Coverdale was inducted into the Rock and Roll Hall of Fame as a member
of Deep Purple, giving one of the bandś induction speeches...
Response: Deep Purple
-------------------------------------------------------------------
Ground Truth Answer: Deep Purple
```

Figure 7: **Conditional embeddings correctly returns the golden document as the first document (and therefore most similar/informative to answer user question) in the set of retrieved documents.** Without conditional embeddings, the golden document is present near the end of the set of retrieved documents, making it harder for the LLM to effectively utilize the relevant context as there is an implicit bias towards tokens present at the beginning of context (Liu et al., 2023b)

## A.2 QUERY REWRITING EXAMPLES

We include empirical examples highlighting observed benefits of iterative query rewriting in fig. 8, fig. 9, and fig. 10.

```
Question: "Call Me Mañana" is a song by a group that has sold how
many records?

Without Query Rewrite:
Response:Two million records

With Query Rewrite:
Rewritten Query: How many records has the German techno group
Scooter sold?
Response: Over 30 million
-------------------------------------------------------------------
Ground Truth Answer: Over 30 million
```

Figure 8: **Query rewriting allows for the LLM to incorporate relevant pre-training data with provided context.** The retrieved documents in both scenarios have no occurrence of 'Scooter', meaning this information must have been present in pre-training and was accessed through query rewriting.

## A.3 QuALITY RESULTS

We observe significant accuracy gains in table 6 when utilizing conditional embeddings on the QuALITY dataset. We do not exclude questions tagged as 'difficult' (questions where less than 50% of the speed annotations by human evaluators were correct) in our evaluation (Pang et al., 2022).

```
Question: What was the 76ers' record the year Allen Iverson won MVP?

Without Query Rewrite:
Response: 71-71

With Query Rewrite:
Rewritten Query: What was the Philadelphia 76ers' record the year
Allen Iverson won the MVP award?
Response: 56-26
-----------------------------------------------------------------------
Ground Truth Answer: 56-26
```

Figure 9: Query rewriting allows for the LLM to parse context with higher efficacy by appending relevant context to the query, aiding in retrieval. **Conditional embeddings (Filtering only) hallucinates a response while query rewriting appends the fact that the 76ers' team is from Philadelphia, facilitating the retrieval of golden documents and generation of the correct answer.**

```
Question: Do you know how many people are in the weeknd?

Without Query Rewrite:
Response: No

With Query Rewrite:
Rewritten Query: How much is The Weeknd's net worth?
Response: The Weeknd is a solo artist, not a band, so there is only
one person in The Weeknd.
-----------------------------------------------------------------------
Ground Truth Answer: 1
```

Figure 10: The rewritten query is unrelated to the question yet it allowed the LLM to determine that the Weeknd is an individual from the documents retrieved by the question about his net worth. **The search during retrieval expanded due to the rewritten question to grab relevant documents which can contain "hints" of the answer to the original question.**

A.4 NATURAL QUESTIONS RESULTS

To extend our suite of knowledge-intensive QA tasks, we additionally evaluate TTRAG on the Natural Questions dataset in table 7. Natural Questions includes questions spanning multiple domains and multi-hop questions (answer can only be synthesized by referring to multiple documents), which often poses a challenge for retrieval based systems when coupled with large context sizes.

A.5 PUBMEDQA RESULTS

We additionally present results for PubmedQA in table 8. Although the domain specific context provided per question in PubMedQA can fit within the context window of large context models, filtering the context by relevance to the query leads to higher accuracy of generated responses

A.6 CONDITIONAL EMBEDDINGS IMPLEMENTATION DETAILS

Conditional embeddings contains two distinct components that are present in our experiments: CE (Filtering only) and CE. CE (Filtering Only) applies retrieval over embeddings generated by a pretrained embedding model such as text-embedding-3-small. CE as a whole encapsulates both filtering and generation by generating conditional embeddings over a set of documents by passing them through DistilBERT and then carrying out retrieval over these conditional embeddings. In our experiments, we present CE (Filtering Only) as a benchmark to display the accuracy gains of targeted relevant context in the prompt being superior naively providing a large block of context to the LLM. The flow of CE (Filtering Only) is as follows: generate embeddings of query and documents with pretrained embedding model and run similarity search between query and documents to fetch top k similar documents. CE does the following: apply CE (Filtering Only) first to get a pre-filtered set of

Table 6: Context in QuALITY is a cohesive long block of text, inherently making all context somewhat relevant to a query. **Conditional embeddings clearly delineates fine grained distinctions between documents pertaining to a similar topic and domain, resulting in a 6% improvement over baseline.**

| Model | | QuALITY | |
| --- | --- | --- | --- |
| Name | Embedding Model | Exact Match | F1 Score |
| **GPT-3.5-0125 CE** | **text-embedding-3-small** | **59.57%** | **55.97** |
| GPT-3.5-0125 CE | bm25 | 58.26% | 54.37 |
| GPT-3.5-0125 CE (Filtering only) | text-embedding-3-small | 59.13% | 53.71 |
| GPT-3.5-0125 CE (Filtering only) | bm25 | 55.22% | 52.3 |
| GPT-3.5-0125 Baseline | text-embedding-3-small | 53.48% | 51.20 |
| GPT-3.5-0125 Baseline | bm25 | 52.17% | 48.58 |

Table 7: Conditional embeddings (Filtering only) maintains high performance when handling up to almost 10 million token context lengths, with a **3% improvement over baseline in open source models.**

| Model | | NQA(1M) | | NQA(5M) | | NQA(9.6M) | |
| --- | --- | --- | --- | --- | --- | --- | --- |
| Name | Embedding Model | Exact Match | F1 Score | Exact Match | F1 Score | Exact Match | F1 Score |
| Gemini-1.5-Flash (Baseline) | text-embedding-3-small | 32.00% | 28.25 | 38.05% | 36.03 | 37.18% | 34.32 |
| Gemini-1.5-Pro (Baseline) | text-embedding-3-small | 38.00% | 31.54 | 40.58% | **39.30** | 39.44% | **38.23** |
| GPT-4-0613 Baseline | text-embedding-3-small | 45.65% | 25.59 | 42.27% | 24.70 | 42.90% | 25.03 |
| GPT-4-0613 CE (Filtering only) | text-embedding-3-small | 45.05% | 26.52 | 44.62% | 25.29 | 38.56% | 24.06 |
| GPT-4-0613 CE (Filtering only) | bm25 | 27.93% | 18.83 | 22.90% | 14.88 | 20.51% | 15.17 |
| GPT-4-turbo-2024-04-09 Baseline | text-embedding-3-small | 45.95% | 33.91 | 47.45% | 33.8 | **45.45%** | 31.93 |
| GPT-4-turbo-2024-04-09 CE (Filtering only) | text-embedding-3-small | **47.75%** | **36.89** | **47.83%** | 35.01 | 43.53% | 33.51 |
| GPT-4-turbo-2024-04-09 CE (Filtering only) | bm25 | 39.64% | 32.15 | 36.26% | 28.6 | 34.52% | 25.25 |
| Llama-2-7b-chat-hf-Baseline | text-embedding-3-small | 27.08% | 13.55 | 22.58% | 13.75 | 21.18% | 14.09 |
| **Llama-2-7b-chat-hf CE (Filtering only)** | **text-embedding-3-small** | **30.21%** | **17.02** | **25.43%** | **17.11** | **24.47%** | **16.19** |
| Llama-2-7b-chat-hf CE (Filtering only) | bm25 | 27.08% | 16.13 | 15.37% | 11.86 | 14.12% | 11.67 |
| Claude-3-Haiku-Baseline | text-embedding-3-small | 38.74% | 20.91 | 40.31% | 22.65 | 37.77% | 21.8 |
| **Claude-3-Haiku CE (Filtering only)** | **text-embedding-3-small** | **45.05%** | **33.65** | **39.92%** | **32.98** | **40.34%** | **30.64** |
| Claude-3-Haiku CE (Filtering only) | bm25 | 35.24% | 23.90 | 27.98% | 21.03 | 27.40% | 18.89 |

k documents, generate conditional embeddings for query and each document in this pre-filtered set by passing through DistilBERT base model (uncased version), extracting last hidden state, masking tokens, and averaging across sequence length dimension, and then run similarity search between query and documents to fetch top similar documents among the pre-filtered set of documents. The value-add of conditional embeddings is in capturing fine-grained semantic information. Therefore this application of conditional embeddings is highly performant as we always conditionally embed on k documents, which introduces very minimal latency and memory and also highlights that CE delivers on capturing detailed semantics. Because each document retrieved in the top k set is highly semantically related to the query, the observed gains from applying CE on this pre-filtered set imply that CE is able to differentiate more detailed, relevant context within a set of documents that are all initially viewed as related to the query.

## A.7 EXACT SUBSTRING MATCHING

To objectively evaluate the performance of our Test-time RAG pipeline, we employ the **Exact Match Score** as one of our primary evaluation metrics. This metric measures the precision of the model's output in replicating the ground truth answers, which is critical for assessing the effectiveness of the retrieval and generation process in producing correct answers. We first normalize the model output and the ground truth answers by processing the text to be lowercase and removing unnecessary spacing and characters. After normalization, the exact match score is computed as follows:

- The normalized output is compared to the normalized ground truth. An exact match occurs if the normalized strings are identical or one is contained in an other, reflecting that the response generated by the model precisely matches the expected answer in content.
- The score is calculated by dividing the number of correct predictions (where the normalized output matches the normalized ground truth exactly) by the total number of samples evalu-

Table 8: **In settings where context is long yet constrained to a specific domain, test time filtering still outperforms naively stuffing the prompt with full context.** GPT-4-Turbo has a 128k length context window, which easily accommodates the 5k context length for PubMedQAquestions. However, conditional embeddings (Filtering only) leads to a **3.7% lift in performance over baseline GPT-4-Turbo.**

| Model | | PubMedQA | |
| --- | --- | --- | --- |
| Name | Embedding Model | Exact Match | F1 Score |
| GPT-4-turbo-2024-04-09 Baseline | text-embedding-3-small | 68.50% | 5.42 |
| **GPT-4-turbo-2024-04-09 CE (Filtering only)** | **text-embedding-3-small** | **72.20%** | **4.75** |
| GPT-4-turbo-2024-04-09 CE (Filtering only) | bm25 | 70.10% | 4.74 |
| Claude-3-Haiku-Baseline | text-embedding-3-small | 52.80% | 2.85 |
| **Claude-3-Haiku CE (Filtering only)** | **text-embedding-3-small** | **62.50%** | **37.22** |
| Claude-3-Haiku CE (Filtering only) | bm25 | 57.40% | 32.67 |

ated. This fraction is then expressed as a percentage to represent the proportion of responses that were perfect matches to the ground truths.

## A.8 QUERY REWRITING PROMPT

"Rewrite the user question by adding information provided in the documents to the rewritten question. Return the answer only if it is found explicitly in the provided documents. Otherwise, return only the rewritten question.
User Question: {query}
Documents: {top three}"

## A.9 COST CALCULATIONS

$$\text{Cost} = \text{Input Token Cost} + \text{Output Token Cost}$$

$$\text{Input Token Cost} = \text{Cost per million input tokens} \times \left( \frac{\text{Number of Questions} \times \text{Average Number of Tokens per Question}}{1000000} \right)$$

$$\text{Output token cost} = \text{Cost per million output tokens} \times \left( \frac{\text{Number of Questions} \times \text{Max Number of Output Tokens per Response}}{1000000} \right)$$

## A.10 COST EFFECTIVE

Another implication of our pipeline is its cost effectiveness. One implicit feature of Test-Time RAG is that context processing is targeted as only relevant chunks are shown to the LLM, especially for Question-Answering tasks. As a result, the average amount of tokens processed per generation is significantly less than passing in large blocks of context. Because LLM API providers charge usage based on tokens in and tokens out, large context tasks can incur compounding charges very quickly. This is mitigated by the reduced token processing done in at all components of Test-Time RAG. We display example costs for running our evaluations in 9, and demonstrate that Test-Time RAG attains superior performance at a fraction of the cost.

Table 9: **We demonstrate superior performance with all collective benefits of Test-Time-RAG for 2.8x less cost** when compared to naively running baselines on these datasets.

| Dataset | GPT-4-0613 Baseline | Test-Time RAG w/ GPT-4-0613 |
| --- | --- | --- |
| HotpotQA | $1890.94 | $558.04 |
| Natural Questions | $417.77 | $147.83 |
| QASPER | $146.34 | $66.42 |

## A.11 ADDITIONAL INFORMATION - CONDITIONAL COMPUTE

For space considerations, we have included performance of conditional compute in this appendix section. We additionally present performance of conditional compute on the KV retrieval task in fig. 13, summarization task in fig. 11 and display aggregate performance averaged across all conditional compute tasks in fig. 12

Table 10: **Conditional compute has lower latency and processes a lower order of magnitude of input tokens when tackling KV retrieval queries**.

| | Latency (s) | | | Input Tokens | | | Output Tokens | | |
|---|---|---|---|---|---|---|---|---|---|
| | Mean | P95 | P99 | Mean | P95 | P99 | Mean | P95 | P99 |
| **Conditional Compute** | **7.29** | 8.01 | 8.75 | **227.67** | 239.0 | 241.0 | **23.76** | 28.0 | 30.0 |
| Baseline | 7.87 | 9.52 | 13.79 | 6987.0 | 7028.0 | 7039.02 | 24.45 | 28.0 | 29.0 |

Table 11: **Baseline methods for summarization process 5x more tokens and have comparable latency to the iterative summarization process employed by conditional compute.**.

| | Latency (s) | | | Input Tokens | | | Output Tokens | | |
|---|---|---|---|---|---|---|---|---|---|
| | Mean | P95 | P99 | Mean | P95 | P99 | Mean | P95 | P99 |
| **Conditional Compute** | 31.07 | 55.21 | 71.62 | **20792.37** | 21184.35 | 21550.19 | **785.37** | 1177.35 | 1543.19 |
| Baseline | 24.87 | 50.12 | 54.66 | 171500 | 172692.8 | 173505.76 | 1816.8 | 1985.8 | 1998.76 |

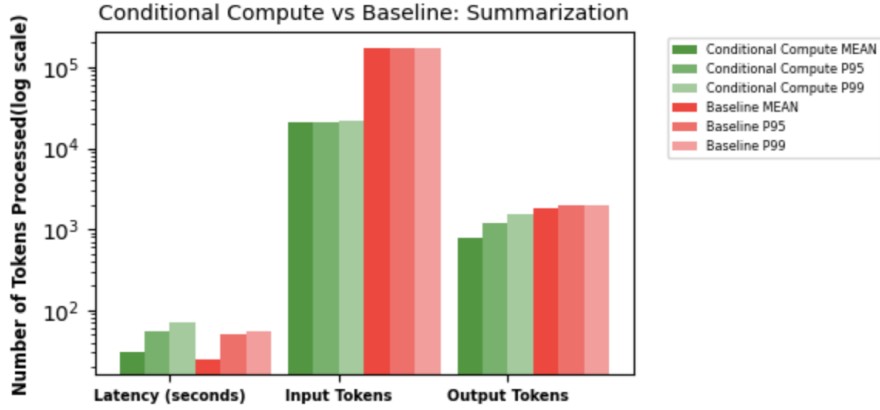

Figure 11: Baseline methods for summarization process 5x more tokens and have comparable latency to the iterative summarization process employed by conditional compute.

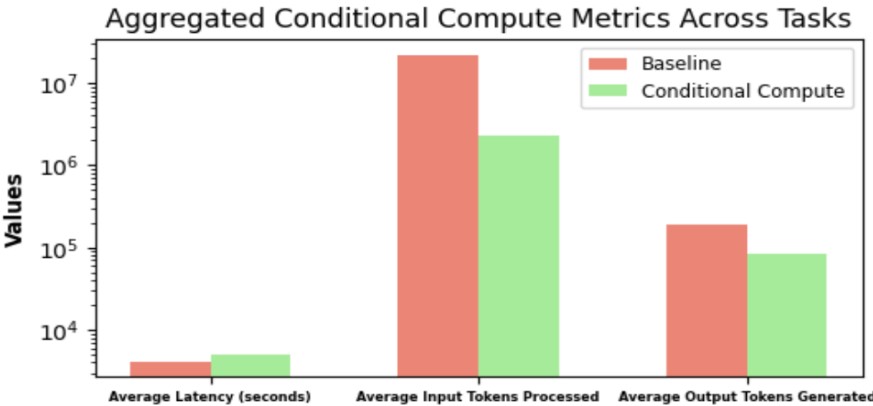

Figure 12: Aggregating key metrics across the different conditional compute tasks, we see that on average, baseline methods process more tokens by a full order of magnitude, which becomes a constraint at inference time in compute and performance for long context tasks.

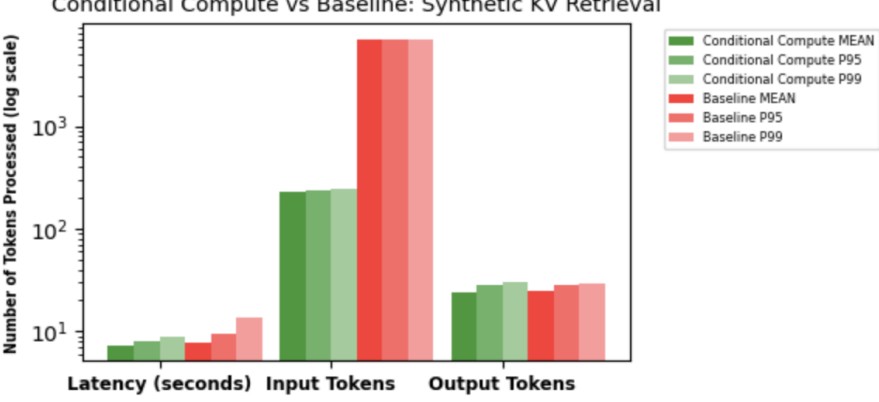

Figure 13: **The number of input tokens processed on KV retrieval tasks by conditional compute is almost an order of magnitude less than baseline**. Additionally, the entire conditional compute pipeline maintains lower latency.

