# OpenReview forum: "Test-Time RAG: Enhancing Long Context Understanding in LLMs with Retrieval-Augmented Mechanisms"
_ICLR.cc/2025/Conference — Submitted to ICLR 2025_

### Official Review · Reviewer_N9d8 · 2024-10-16

**Soundness:** 2
**Presentation:** 1
**Contribution:** 1
**Rating:** 3
**Confidence:** 4

**Summary:**

This paper propose test-time RAG, aiming at constructing more contextual text embeddings to improve the retrieval accuracy. This method consists of 3 technologies: conditional embedding, query rewriting and conditional computing. It conducts experiments on 3 RAG datasets with 2 different LLMs and 2 different embedding models, to demonstrate this method can improve models’ performance.

**Strengths:**

1. The author conducts experiments on 3 RAG datasets with 2 different LLMs and 2 different embedding models to demonstrate the advantage of test-time RAG.

**Weaknesses:**

1. Poor writing

The author’s presentation is hard to understand. I cannot understand the description of the methods even after reading 10 times, such as the sentence “utilize encoder-decoder variants which jointly condition context on text appearing before and after a given token in a sequence” (line 164).

2. Impracticable time consumption

“Conditional embedding” needs to encode each document in test-time, which will certainly take extremely much more time. Even though the author proposes “conditional computing” to reduce the latency, it can only function on a narrow range of task types such as counting or KV retrieval. But the author did not compare the time latency with that of standard RAG methods on more general problems. This casts doubt on the practical feasibility of this approach in real-world scenarios.


3. Query rewriting may introduce more issues

The author claims that query rewriting can leverage knowledge in LLM’s pretraining data. However, it can be affected by LLM’s hallucination, which is exactly what RAG technology wants to solve. Therefore, in most RAG scenarios, LLMs are allowed to only use knowledge in the retrieved documents instead of their inner knowledge.

4. The resolution of the figures is low. And the sizes and positions of them are also inapt.

**Questions:**

Can you test the time consumption of test-time RAG, compared with standard RAG methods, on these general problems?

---

> ### Author Response · Authors · 2024-11-26
> **Rebuttal by Authors**
>
> We’d like to thank the reviewer for their detailed questions and feedback. We’ve addressed questions and concerns below.
>
> * **(Question #1)** For each ablation and experiment, we include baselines which encapsulate two primary data points to benchmark TTRAG against. Our baseline simulates naively stuffing the full context window with information to test the model’s standalone capabilities for processing long context, while CE (Filtering only) effectively simulates traditional search and retrieval methods for selecting relevant context and including that in the prompt. We demonstrate in our current set of experiments that TTRAG as a whole outperforms solely CE (Filtering only), baselines, and other individual components of TTRAG on relevant benchmarks. For clarity, we refrain from calling CE (Filtering only) as standard RAG because we assume apriori a test-time availability of context. Therefore, we cannot pre-index context in the traditional, standard RAG paradigm. With respect to time consumption, we present results below demonstrating comparable latency between traditional RAG paradigms and the components of TTRAG  (on HotpotQA with 1 million token context):
> 1. Conditional Embeddings vs Traditional RAG
> | model | average latency (s) | 95th percentile (s) | 99th percentile (s) |
> |-------|---------------------|-----------------|-----------------|
> | GPT-3.5-Turbo (with Conditional Embeddings) | 1.283 | 1.876 | 3.742 |
> | Llama-2-7B (with Conditional Embeddings) | 3.753 | 6.327 | 11.963 |
> | GPT-3.5-Turbo (Traditional RAG) | 1.076 | 1.539 | 3.315 |
> | Llama-2-7B (Traditional RAG) |  3.339 | 4.941 | 10.628|
> 2. Query Rewriting vs. Without Query Rewriting
> | model | accuracy (%) | average latency (s) | 95th percentile (s) | 99th percentile (s) |
> |-------|------------|---------------------|-----------------|-----------------|
> | GPT-3.5-Turbo (with Query Rewriting) | 45.376 | 3.208 | 4.826 | 7.702 |
> | GPT-3.5-Turbo (without Query Rewriting)| 41.181 |1.283 | 1.876 | 3.742 |
>
> * **(Weakness #1)** We have improved writing clarity in our most recent revision of our paper. With respect to the specific line highlighted, we are highlighting that a core difference between encoder-decoder models and decoder-only models is the application of attention to tokens, where bidirectional attention effectively creates token representations that encapsulate contextual information from tokens earlier in the sequence and later in the sequence, as opposed to masked attention which is applied only to prior tokens in the sequence.
> * **(Weakness #2)** To clarify, for performance considerations in all of our experiments, we first retrieve the top 10 documents using a generic embedding model like text-embedding-3-small, and then conditionally embed these top 10 documents with the query and retrieve a smaller subset of relevant documents from this ‘pre-filtered’ set. The value-add of conditional embeddings is in capturing fine-grained semantic information, so this application of conditional embeddings accomplishes 2 things: (1) It is highly performant as we always conditionally embed on 10 documents, which introduces very minimal latency and memory. (2) Highlights that CE delivers on capturing detailed semantics. Because each document retrieved in the top 10 set is highly semantically  related to the query, the observed gains from applying CE on this pre-filtered set imply that CE is able to differentiate more detailed, relevant context within a set of documents that are all initially viewed as related to the query.  Conditional embeddings boasts comparable time consumption when compared to traditional RAG methods, while leading to accuracy gains (please refer to our response for 'Question #1' for this data).
> * **(Weakness #3)**  Runaway results for query rewriting are prevented in two ways: First, in the case that the answer is not found in the retrieved documents after all iterations of rewrites (number of iterations until stopping is fixed to 3 in our experiments but this can be treated as a hyperparameter), query rewriting defaults to just fetching the top k (another hyperparameter fixed to 10) documents from similarity search with the original un-rewritten query, lower bounding the performance of query rewriting with normal retrieval. If the answer is found and we want to use the retrieved documents from the rewritten query, at the answer generation stage, retrieved documents from the new query are passed in along with the original query. This prevents the LLM from generating irrelevant responses while maintaining access to information found with the rewritten query.
> * **(Weakness #4)** We have addressed these concerns surrounding figure display in our revised iteration of our paper.

---

> > ### Author Response · Authors · 2024-11-28
> >
> > Dear Reviewer,
> >
> > Thank you again for your insightful review. It has greatly improved our paper, particularly with improving the writing mechanics describing conditional embeddings and query rewriting as well as running ablations that highlight the low-latency of our proposed TTRAG approach.
> > As the public discussion phase is nearing its conclusion, we wanted to follow up to see if there are any additional questions, concerns, or points that we could clarify or address to further assist the your review. We are more than happy to provide any additional experiments, ablations or clarifications you might need.
> >
> > Thank you!

---

### Official Review · Reviewer_beNT · 2024-10-24

**Soundness:** 2
**Presentation:** 2
**Contribution:** 2
**Rating:** 3
**Confidence:** 4

**Summary:**

The paper introduces Test-time RAG, a framework that integrates Retrieval-Augmented Generation (RAG) with dynamic query handling at test time. It aims to improve the long-context understanding of Large Language Models (LLMs) by addressing the limitations of static retrieval and pre-indexed document embeddings. The framework consists of three main components: Conditional Embeddings, Iterative Query Rewriting, and Conditional Compute, which work together to enhance context processing dynamically. The paper evaluates Test-time RAG on several benchmarks, showing improved performance in handling complex, long-context retrieval tasks such as knowledge-intensive question answering (QA).

**Strengths:**

The starting idea is insteresting: dealing with streaming and changing long context with tailored RAG.

**Weaknesses:**

1. **Poor Writing and Organization**: The presentation of the paper is confusing and difficult to follow due to ambiguous and hard-to-read sentences, poor organization, and unclear structuring of key arguments.

2. **Inconsistent Focus**: The introduction starts by discussing personalized and trustworthy AI, but the subsequent sections focus on long-context benchmarks without sufficiently connecting to the original arguments.

3. **Strange Motivation for Test-time Processing**: The rationale for processing context at test time appears weak, especially given that user history or dynamically changing contexts could be indexed incrementally.

4. **Lack of Component Integration**: The paper presents three distinct components (Conditional Embeddings, Iterative Query Rewriting, Conditional Compute) but does not adequately explain their interrelationship. The experiments are also conducted individually rather than for the overall method, which leaves the effectiveness of the integrated system unclear.

5. **Inefficient Indexing**: The experiments on Conditional Embedding involve incorporating the query into the document indexing process, which makes it impractical because the indexed embeddings cannot be reused efficiently.

6. **Weak Experimental Validation**: The experiments provided are not comprehensive or robust enough to fully validate the effectiveness of the proposed techniques.

7. **Lack of Discussion or Justification**: There is minimal discussion on why or how the proposed techniques improve the results, leaving the reader with questions about the validity and significance of the approach.

**Questions:**

1.	What is the specific connection between the initial focus on personalized and trustworthy AI and the later focus on long-context benchmarks? Could the authors clarify the relevance of these two aspects?
	2.	Why was the decision made to process context at test time when many changing contexts (e.g., user history) could be indexed incrementally? What benefits does test-time processing offer in these cases?
	3.	How are the three components (Conditional Embeddings, Iterative Query Rewriting, Conditional Compute) related to each other? Could the authors provide a clearer description of how these components work together within the overall system?
	4.	Why were most experiments focused on Conditional Embeddings rather than evaluating the full method? What impact do Iterative Query Rewriting and Conditional Compute have when used together with Conditional Embeddings?

---

> ### Author Response · Authors · 2024-11-26
> **Rebuttal by Authors**
>
> We’d like to thank the reviewer for their constructive feedback and time. We address them below.
>
> * **(Question #1)** An example to demonstrate the connection between personalized AI and long-context is the following: Consider receiving an email with sensitive information from a coworker, with a large attachment of many pages. Generating a response to this email involves the following considerations: 1) long-context processing and understanding of the email and its attachments 2) an inability to pre-index the attachment, because it was just received and we need to understand it on the fly and it contains private information. Use-cases like this highlight the importance of processing context at test-time. This builds motivation for the rest of the paper, as one unique quality of test-time processing, is that we have simultaneous access to user intent (query) and relevant context, which we then exploit via conditional embeddings, query rewriting, and conditional compute. The long-context benchmarks demonstrate that these techniques perform well at scale (to tie it back to the example: in cases where the attachment in the email is millions of tokens).
> * **(Question #2)** Although not mutually exclusive with incremental indexing, TTRAG focuses on scenarios where we do not have the ability to evaluate and retain the embeddings ahead of time.
> * **(Question #3 and #4)** We have updated our paper and methodology to discuss the complete flow in test time RAG and include results for the collective benefits of the entire pipeline when stitched together. Please refer to table 1, and an updated figure 1 which illustrates how TTRAG works as a comprehensive system. For a brief summary (assume a question-answering setting):
> 1. We receive a block of context and question.
> 2. Conditional compute routes the query to one of the compute categories (in this case question-answering)
> 3. Generate conditional embeddings, select top k documents
> 4. Use query rewriting to rewrite query if answer isn’t found
> 5. Re-generate conditional embeddings with rewritten query
> 6. Repeat steps 4 and 5 until stopping criterion for total rewrites is hit or answer is found (whichever comes first)
> 7. Generate response
>
> * **(Weakness #6)** We have modified the paper to demonstrate our experiments from a ‘top-down’ approach, where we start by presenting the performance gains of TTRAG as a whole, then ablating on individual components to demonstrate their individual contributions to performance. Additionally, for each ablation and experiment we include baselines which encapsulate two primary data points to benchmark TTRAG against. Our baseline simulates naively stuffing the prompt with information, while CE (Filtering only) effectively encapsulates traditional search and retrieval methods for selecting relevant context. We demonstrate in our benchmarks that TTRAG outperforms these baselines.

---

> > ### Comment · Reviewer_beNT · 2024-11-26
> >
> > Thank you for the authors' response. After reading the revised paper, I believe that the current version still falls short of the acceptance criteria. While the authors' response addresses some of my concerns, several issues remain unresolved. I have adjusted my assessment accordingly to reflect the updates.

---

> > > ### Author Response · Authors · 2024-11-28
> > >
> > > Dear Reviewer,
> > >
> > > Thank you again for your insightful review. It has greatly improved our paper, particularly with bolstering our experiments with results highlighting the significant accuracy improvements from TTRAG as a whole and clarifying how each component fits together in the pipeline to substantiate these gains.
> > > As the public discussion phase is nearing its conclusion, we wanted to follow up to see if there are any additional questions, concerns, or points that we could clarify or address to further assist the your review. We are more than happy to provide any additional experiments, ablations or clarifications you might need.
> > >
> > > Thank you!

---

### Official Review · Reviewer_WN1v · 2024-10-28

**Soundness:** 2
**Presentation:** 2
**Contribution:** 2
**Rating:** 5
**Confidence:** 4

**Summary:**

The paper looks at the problem of long-context RAG and provides 3 steps to solve it. Their overall approach, Test-Time RAG considers a custom embedding approach called Conditional Embeddings, uses query rewriting and then use conditional compute. They evaluate their results on a few different datasets showing consistent improvement in results over baseline.

**Strengths:**

The paper handles an important problem around long-context RAG. Their approach of using a conditional compute is important and shows the versatility of tasks in a question answering system and how a LLM prompt may not be the best way to do all queries. Query rewriting based on iterative search is an interesting idea too. The problem has practical implications in both research and industry.

**Weaknesses:**

The paper presents an interesting new perspective on long context retrieval. However the work has some limitations worth considering

1.  Section 3.1 introduces conditional embeddings however the authors do not state they are using pretrained embeddings for the same in this section. It reads like this is a custom embedding. Much later, in Table 1 we understand that the authors have used GPT3.5 embeddings and llama embeddings for the same. This makes reading section 3.1 hard without forward referencing
2. It is not clear why an average across the unmasked tokens in CE embeddings is better than taking a  weighted average (with learned weights)
3. Scalability of solution - how does this solution scale with respect to large datasets of documents ?
4. I do not understand on how you prevent Algorithm 1 to create irrelevant questions if the retrieval is of poor quality. The approach runs the risk of run away poor results on questions which do not have good answers in the database
5. Lack of clarity in what is CE and what is "CE (Filtering only)" in Table 1.
6. Lack of clarify that the model is a LLM model used only for question answering
7. How is F1 score computed? Over how many documents i.e. what is your retrieved k?
8. Whilst I appreciate the principle behind the conditional compute, it's relevance to the rest of the paper is relatively weak. Also how is the LLM leverage For example for count-based queries the map-reduce paradigm that has been recommended, does it mean the LLM generates the code for map-reduce? if not how will this be implemented?
9. The example in Figure 6 is a good example of my earlier comment on Conditional Rewrite. If the "Call me Manana" is not present in the document repository the query rewrite may tag it to some random group (based on what is retrieved) in the way the algorithm is designed/presented. it is not clear how this run-away is prevented to attach it to something random in what has been retrieved. Similar comment on fig 7/8


Overall, while interesting the paper has many unanswered questions. First, it is not clear on the cost vs. benefit analysis for the CE approach. Second, the conditional rewrite has a risk of runaway poor results which need to be addressed. Finally conditional compute is explained but how are some of it implemented is vague. In view of this, this paper in the current form leaves many questions unanswered on its applicability. More clarity on writing is preferable as there is a need to read further to understand the methodology.

**Questions:**

1. What is the architecture used in the bidirectional encoder? How do the authors arrive at this architecture?
2. Please explain why average of unmasked tokens is better than a weighted sum of the unmasked tokens where the weights can be learned? either a theoretical explanation or an empirical study is preferred.
3. Clearly differentiate between CE vs CE Filtering only
4. What are the performance implications of Test-Time RAG? On the infrastructure which you are using (which is not specified?) how much time and GPU memory does CE introduce? Given the gains in Tables 1 and 2 are relatively less, is it worth the compute?
5. Can statistical tests (even a means test) be done on each of the groups of 3 in tables 1 and 3 highlighting which numbers are statistically better than the baseline and where is CE better than CE Filtering?
6. What are the time impacts on iterative query rewriting ? What is the distribution of the number of rewrites as per the algorithm? how long does the response take with rewrites vs. without on average?

---

> ### Author Response · Authors · 2024-11-26
> **Rebuttal by Authors**
>
> We would like to first thank the reviewer for their detailed feedback and insightful questions. We have addressed questions and weaknesses below.
>
> * **(Question #1)** We utilize DistillBERT and extract the last hidden state of the encoder portion. We do not introduce a new architecture for generating conditional embedding. We chose to utilize the bidirectional encoder present in BERT because each token can update its representation based on all tokens in the sequence (whether they appear before or after it in the sequence). This advantage is not present if we just extracted the last hidden state of a decoder-only model like Llama-2.
> * **(Question #2)** Averaging is done primarily because at test-time, we don’t have access to information to determine a learned weightage of tokens. We assume we receive the tokens at test time, therefore it is not possible in this setting to learn a weightage on the fly for token ‘importance’. Because we don’t know the prior significance of each token  of context, this necessitates taking an average of unmasked tokens.
> * **(Question #3)** CE Filtering Only applies retrieval over embeddings generated by a pretrained embedding model. CE as a whole generates conditional embeddings and then carries out retrieval over these conditional embeddings (no pretrained embedding model). The flow of CE Filtering Only is as follows: generate embeddings of query and documents with pretrained embedding model (we utilize text-embedding-3-small) and run similarity search between query and documents to fetch top k similar documents. CE as a whole does the following: apply CE Filtering Only first to get a pre-filtered set of 10 documents, generate conditional embeddings for query and each document in this pre-filtered set by passing through bidirectional encoder, extracting last hidden state, masking tokens, and averaging across sequence length dimension, and then run similarity search between query and documents to fetch top similar documents amongst the pre-filtered set of documents.
> * **(Question #4)** To clarify, for performance considerations in all of our experiments, we first retrieve the top 10 documents using a generic embedding model like text-embedding-3-small, and then conditionally embed these top 10 documents with the query and retrieve a smaller subset of relevant documents from this ‘pre-filtered’ set. The value-add of conditional embeddings is in capturing fine-grained semantic information, so this application of conditional embeddings accomplishes 2 things: (1) It is highly performant as we always conditionally embed on 10 documents, which introduces very minimal latency and memory. (2) Highlights that CE delivers on capturing detailed semantics. Because each document retrieved in the top 10 set is highly semantically  related to the query, the observed gains from applying CE on this pre-filtered set imply that CE is able to differentiate more detailed, relevant context within a set of documents that are all initially viewed as related to the query.  Conditional embeddings boasts comparable time consumption when compared to traditional RAG methods, while leading to accuracy gains. We present empirical results below for the time consumption of conditional embeddings below (on HotpotQA with 1 million token context), which demonstrate comparable latencies between traditional RAG methods and conditional embeddings:
> 1. Conditional Embeddings vs Traditional RAG
> | model | average latency (s) | 95th percentile (s) | 99th percentile (s) |
> |-------|---------------------|-----------------|-----------------|
> | GPT-3.5-Turbo (with Conditional Embeddings) | 1.283 | 1.876 | 3.742 |
> | Llama-2-7B (with Conditional Embeddings) | 3.753 | 6.327 | 11.963 |
> | GPT-3.5-Turbo (Traditional RAG) | 1.076 | 1.539 | 3.315 |
> | Llama-2-7B (Traditional RAG) |  3.339 | 4.941 | 10.628|
>
> * **(Question #6)** We present an empirical result displaying the latency of query rewriting. Although there is an increase in latency, we observe substantial accuracy gains.
> 1. Query Rewriting vs Without Query Rewriting
> | model | accuracy (%) | average latency (s) | 95th percentile (s) | 99th percentile (s) |
> |-------|------------|---------------------|-----------------|-----------------|
> | GPT-3.5-Turbo (with Query Rewriting) | 45.376 | 3.208 | 4.826 | 7.702 |
> | GPT-3.5-Turbo (without Query Rewriting)| 41.181 |1.283 | 1.876 | 3.742 |

---

> > ### Author Response · Authors · 2024-11-26
> > **Rebuttal by Authors (continued)**
> >
> > * **(Weakness #3)** In our paper, Table 1 (1 million and 5 million tokens ), Table 3 (1 million - 8.8 million tokens), Table 4 (330k tokens),  and Table 7 (1 million - 9.6 million tokens) demonstrate TTRAG’s performance at very large context sizes. One of TTRAG’s strengths is that it performantly scales to large document sets. The experiments in the aforementioned tables contain large sets of documents up to 9.6 million tokens and we empirically demonstrate consistent performance across varying context lengths at a large scale.
> > * **(Weakness #7)** In our experiments, we fix the retrieved k to 10 documents and each document has a  chunk size of 200 tokens. F1 score is computed by the standard definition: $$\frac{2 * Precision * Recall}{Precision + Recall}$$
> > * **(Weakness #8)** We have updated the writing mechanics in our paper for more clarity. The LLM does not generate any code during conditional compute. The LLM router’s sole responsibility is to route the problem to the discrete set of categories we have enumerated. We have implemented efficient methods for solving each ‘compute category’ manually.
> > * **(Weakness #4 and #9)** Runaway results for query rewriting are exhaustively prevented in 2 distinct cases: when the answer is not found during the query rewriting process and when the answer is found. First, in the case that the answer is not found in the retrieved documents after all iterations of rewrites (number of iterations until stopping is fixed to 3 in our experiments but this can be treated as a hyperparameter), query rewriting defaults to just fetching the top k (another hyperparameter fixed to 10) documents from similarity search with the original un-rewritten query, lower bounding the performance of query rewriting with normal retrieval. In the case that the answer is found and we want to use the retrieved documents from the rewritten query, at the answer generation stage, retrieved documents from the new query are passed in along with the original query. This prevents the LLM from generating irrelevant responses while maintaining access to information found with the rewritten query.

---

> > > ### Comment · Reviewer_WN1v · 2024-11-26
> > >
> > > Thank you for your detailed response. I will go over it by tomorrow and provide detailed feedback and if scores can be revised. One question - are you updating the document especially with clarifications on CE vs CE (filtering only) and on conditional embeddings? If so please let know by when and how do we find updates .

---

> > > > ### Author Response · Authors · 2024-11-27
> > > >
> > > > Our most recent version of the paper contains clearer elaborations between CE and CE (Filtering only) in section 3.1 (starting at line 172). We hope that this section along with our rebuttals for Question #3 and #4 provide clarity on this topic. Thank you for your reply and feedback.

---

> > > > > ### Comment · Reviewer_WN1v · 2024-11-27
> > > > >
> > > > > Thank you for your detailed response to the comments and the updated paper. Would have appreciated if the updates were in a different colour as some papers have done but it is fine.
> > > > >
> > > > > Now coming to specifics to your comments:-
> > > > > 1. The details on DistilBert are still not mentioned in your updated manuscript (searched for bert and distil)
> > > > > 2. I understand you will get tokens at test time but I am not fully clear why this cannot be incorporated in your training paradigm to learn trained weights. Maybe I am missing something still.
> > > > > 3. Thanks for the clarification on CE vs CE (Filtering only). This clarification however is not present in the manuscript and it is hard to follow even now from the manuscript alone what is the difference between the two
> > > > > 4. On Q4 and Q6 thank you for your experimental details. It is not clear to me why CE will have improved latency  over RAG.
> > > > >
> > > > > On the comments on weaknesses, I cannot find even in the updated manuscript the detail on recovering 10 documents. Let me know where this is present in the current version of the document. Without this, I am not sure how can one understand F1 scores. Also do questions have potentially multiple correct documents or is there always one correct answer to a question in your train/test datasets. Are there examples of so-called "impossible" questions in your data? All of this makes interpretation of F1 score (whose formula I was aware of, but thanks) very difficult.
> > > > >
> > > > > For the points on conditional compute, again I do not see the manuscript to explain the details on what is being generated. Also how the rerouting is happening and details on "We have implemented efficient methods for solving each ‘compute category’ manually" are missing.
> > > > >
> > > > > In summary, the authors have provided detailed responses and would like to acknowledge the same. However, like I have pointed above not everything is still reflecting in the manuscript (not even in appendices). The paper is still hard to read and figure out and would need at least a minimum of referring to these comments to grasp, that too difficultly.
> > > > >
> > > > > In view of the same and my detailed observations, I would like to keep my scores unchanged.

---

> > > > > > ### Author Response · Authors · 2024-11-30
> > > > > >
> > > > > > Dear Reviewer,
> > > > > >
> > > > > > Thank you for your detailed response. We’ve updated our manuscript with the following additions for clarity:
> > > > > > 1. Specifying DistilBERT as the base model for generating conditional embeddings (lines 192, 805, 841)
> > > > > > 2. Explicit clarification for CE vs CE (Filtering only) in the appendix section A6 (“Conditional Embeddings Implementation Details” starting at line 800)
> > > > > >
> > > > > > With respect to your other comments and concerns:
> > > > > > 1. We discuss the details of the implementation for efficient methods used in conditional compute in section 3.3 (starting at line 261). In summary, we pass the question along with the 4 compute categories (KV retrieval, summarization, aggregate statistics, and question-answering) and prompt an out-of-the-box LLM (GPT-3.5-turbo) to determine which category the question falls into. This is the 'routing' step used to determine the conditional compute category. Subsequently, the corresponding conditional compute method is utilized.
> > > > > > 2. In response to “Question #2,” we’d like to clarify that there is no training paradigm present in any component of TTRAG. DistilBERT is used out-of-the-box for conditional embeddings.
> > > > > > 3. The top 10 documents are retrieved via cosine similarity search (details mentioned in lines 199 and 337). We refer to the set of retrieved documents as 'top k' throughout the rest of the paper to emphasize that the number of documents in the retrieved can be set as a hyperparameter. The datasets we utilize contain questions where either a singular golden document is required to answer a question or questions can be of multi-hop nature, which require information spanning multiple documents to generate an answer. The only dataset which contains “impossible questions” where the provided context may not contain an answer to the question is the CRAG dataset [1], which includes these types of questions by design.
> > > > > > 4. CE has a slightly higher latency over traditional RAG but grants higher accuracy (results observed in Table 2/line 406 and Table 6/line 810 in manuscript). Apologies, the results in our previous comment had mistakenly swapped rows between Traditional RAG and CE and we have updated the comment to reflect the correct latencies.
> > > > > >
> > > > > > We’d like to thank you again for your feedback as it has greatly improved our paper, particularly with establishing procedural clarity for conditional embeddings and ablations that highlight TTRAG’s low latency.
> > > > > > As the public discussion phase is nearing its conclusion, we wanted to follow up to see if there are any additional questions, concerns, or points that we could clarify or address to further assist the your review. We are more than happy to provide any additional experiments, ablations or clarifications you might need.
> > > > > > Thanks!
> > > > > >
> > > > > > [1] CRAG dataset: https://www.aicrowd.com/challenges/meta-comprehensive-rag-benchmark-kdd-cup-2024/problems/meta-kdd-cup-24-crag-retrieval-summarization

---

> > > > > > > ### Comment · Reviewer_WN1v · 2024-12-01
> > > > > > >
> > > > > > > Thank you for your response. It is good that CE vs CE filtering only has now been clarified in an appendix. You have explained that k is a hyperparameter for your experiments - whilst that is always the case it should be always be mentioned what value of k are you reporting the results for, are there multiple correct answers to a question, are there more than one correct answer to a question etc. Without these scores become very hard to interpret especially F1-score. A clear definition of exact match is also missing - except in A.7 without (unless I have missed it) a reference to it from the main paper.
> > > > > > >
> > > > > > > I appreciate the time and effort you have put in towards the paper and its rebutall. However, I do not see this paper to be beyond the acceptance limit.  Among other things, the paper needs a thorough rewrite even after the significant changes done during the rebuttal. Would encourage the authors to consider the comments for all reviewers in future versions of this paper.
> > > > > > >
> > > > > > > Once again thank you for your time and effort but apologies I would need to keep my scores unchanged since I am not convinced this is beyond the acceptance threshold.

---

### Official Review · Reviewer_c19t · 2024-11-04

**Soundness:** 2
**Presentation:** 1
**Contribution:** 1
**Rating:** 3
**Confidence:** 4

**Summary:**

The authors propose an iterative query rewriting approach that sequentially and conditionally embed the query with retrieved documents to improve long context understanding.

**Strengths:**

The authors propose to iteratively rewrite the query to improve search results for extensive personalized context, which is a timely problem of interest, with a significant body of literature.

**Weaknesses:**

Overall confused by the claims of the objective of the work and the actual paper body, it is not clear how the work relates to “long context understanding”, “personalised context”, or “specialised task-oriented systems”.  The authors proposed a rewriting query scheme with iterative embedding, with substantial details missing (and therefore, conclusiveness), see below.

- There is a substantial amount of work on query rewriting and agentic RAG which is not mentioned at all. There is no comprehensive literature review, with the “motivation” section being almost unrelated. Not able to compare to other related papers like “Query Rewriting for Retrieval-Augmented Large Language Models”, among many others.
- “Test-time” term is misleading. Similarly, the authors didn’t clearly show benefits of the approach specifically for “long context”, as claimed. The examples do not illustrate long context, only short context.
- Benchmarks do not seem well suited for the specific problem(s) claimed by the authors. Also, BM25 is not an embedding model.
- Motivation for the conditional compute not clear, context with long context understanding missing. Seems ad-hoc and not reproducible.
- A significant number of question arise, which is caused partially by the lack of reference to state of the art methodologies: how does this work compare to reranking methodologies? What is the latency of the pipeline? When does it stop the iteration loop (what are the stop criterion)? What is the effect of these stop criterion? How significant are the small increments in performance claimed by the authors?
- Lack of overall reproducibility for different datasets

**Questions:**

/

---

> ### Author Response · Authors · 2024-11-26
> **Rebuttal by Authors**
>
> We’d like to thank the reviewer for spending time to provide us with valuable feedback. Below, we address your questions and concerns.
>
> *  **(Weakness #1)** We present query rewriting as a component of TTRAG which is enabled by access to both the specific query and context at test-time. A key differentiator between the query rewriting process in TTRAG and the paper highlighted by the reviewer [1] is that query rewriting in TTRAG is driven by an out-of-the-box LLM rewriting the query based on information present in the retrieved documents during each iteration, rather than utilizing a small language model trained via reinforcement learning from LLM feedback [1]. In short our approach does not utilize a training scheme and rewrites queries with information found in documents while [1] utilizes a trainable scheme to do so. Because query rewriting is only a singular component of TTRAG, our motivation section focuses on highlighting current methods of long-context processing (as TTRAG is applied in long context settings). Our  comment addressing ‘Weakness #2’ provides more context and motivation for the connection between personalized AI, long-context, and how it relates to TTRAG.
>
> * **(Weakness #2)** An example to demonstrate the connection between personalized AI and long-context is the following: Consider receiving an email with sensitive information from a coworker, with a large attachment of many pages. Generating a response to this email involves the following considerations: 1) long-context processing and understanding of the email and its attachments 2) an inability to pre-index the attachment, because it was just received, we need to understand it on the fly and it contains private information. Use-cases like this highlight the importance of processing context at test-time. This builds motivation for the rest of the paper, as one unique quality of test-time processing, is that we have simultaneous access to user intent (query) and relevant context, which we then exploit via conditional embeddings, query rewriting, and conditional compute. The long-context benchmarks demonstrate that these techniques perform well at scale (to tie it back to the example: in cases where the attachment in the email is millions of tokens).
>
> * **(Weakness #3)** Within our benchmarks we include long-context datasets spanning 1 million to 9.6 million tokens. We additionally include ablations on datasets with smaller token contexts per question that can fit within the context window of most modern LLMs to demonstrate that our method performs better than naively filling the entire context window of the LLM. We refer to the sparse vector embeddings that can be generated by BM25 when listing BM25 as the ‘embedding model’.
>
> * **(Weakness #4)** The motivation behind conditional compute with respect to long context is that certain questions with long context, such as needle in a haystack tasks, can be determined very quickly without needling to explicitly process the entire block of context which is both costly and time consuming. The majority of questions will fall under question and answering, which defaults to the full TTRAG pipeline but in cases that fall in the compute categories we’ve enumerated, we observe latency gains without losing out on performance. At its core, conditional compute relies on an LLM router which classifies the compute category of a query. If the query falls into a category other than default question answering, a pre-implemented compute method is utilized, else we default to the full TTRAG pipeline. For a visual representation of this in action, please refer to figure 1 in our revised paper.
>
>
> [1] 'Query Rewriting for Retrieval-Augmented Large Language Models' https://arxiv.org/abs/2305.14283

---

> ### Author Response · Authors · 2024-11-26
> **Rebuttal by Authors (continued)**
>
> * **(Weakness #5)** Because the focus of TTRAG is to tackle long-context settings rather than re-ranking specifically, we present experiments and references to compare how well current LLMs can naively process long context at test time with TTRAG. With respect to query rewriting, we outline algorithm specifics including stop criterion for iterations in Algorithm 1 in our revised paper. We establish that stop criterion and the number of documents retrieved per iteration can be treated as hyperparameters. In our experiments, we retrieve the top 3 documents during each rewrite iteration with the stopping criterion to be the minimum of either 3 rewrites or when the answer is found in the top 3 retrieved documents. For latency of the pipeline, we display results for conditional embeddings and query rewriting below (on HotpotQA with 1 million token context), demonstrating comparable latencies between TTRAG components and traditional RAG:
> 1. Conditional Embeddings vs Traditional RAG
> | model | average latency (s) | 95th percentile (s) | 99th percentile (s) |
> |-------|---------------------|-----------------|-----------------|
> | GPT-3.5-Turbo (with Conditional Embeddings) | 1.283 | 1.876 | 3.742 |
> | Llama-2-7B (with Conditional Embeddings) | 3.753 | 6.327 | 11.963 |
> | GPT-3.5-Turbo (Traditional RAG) | 1.076 | 1.539 | 3.315 |
> | Llama-2-7B (Traditional RAG) |  3.339 | 4.941 | 10.628|
> 2. Query Rewriting vs. Without Query Rewriting
> | model | accuracy (%) | average latency (s) | 95th percentile (s) | 99th percentile (s) |
> |-------|------------|---------------------|-----------------|-----------------|
> | GPT-3.5-Turbo (with Query Rewriting) | 45.376 | 3.208 | 4.826 | 7.702 |
> | GPT-3.5-Turbo (without Query Rewriting)| 41.181 |1.283 | 1.876 | 3.742 |
>
>
> * **(Weakness #6)** We outline our methodology for establishing baselines and experiments as well as any auxiliary data preprocessing steps for reproducibility. For reference, there are 2 key considerations to generalize TTRAG experiments to other datasets:
> 1. Creating a large enough knowledge base to simulate retrieval over long context (in datasets where context provided per question is not sufficiently large): This is accomplished by extracting context for across all questions in the dataset, concatenating them together until the context block is large enough, and then ensuring that we only ask questions whose answer is contained within this context block. This ensures that we fairly test model’s generation capabilities as the golden documents are guaranteed to be present in the context.
> 2. Baseline against ‘naively’ filling the context window of current LLMs: The purpose of this baseline is to establish how well a given LLM can utilize information present across a completely filled context window.
>
> Because most current models cannot support multi-million token context lengths, we circumvent this in our experiments by retrieving the top 10 documents (200 tokens per document), then backfilling the context window with randomly retrieved documents from the large context block for the dataset (which is generated in step 1), and then shuffling the documents in the context window. These steps are agnostic to the dataset and are used in our experiments with datasets such as Natural Questions and HotpotQA. For datasets such as QuALITY and QASPER, where substantial context is provided per question, we do not do consideration (1) and just utilize consideration (2) to establish baselines. TTRAG demonstrates performance gains across a variety of datasets with varying structure and type.

---

> ### Author Response · Authors · 2024-11-28
>
> Dear Reviewer,
>
> Thank you again for your insightful review. It has greatly improved our paper, particularly with demonstrating  experimental reproducibility and ablations that highlight the low-latency of our proposed TTRAG approach.
> As the public discussion phase is nearing its conclusion, we wanted to follow up to see if there are any additional questions, concerns, or points that we could clarify or address to further assist the your review. We are more than happy to provide any additional experiments, ablations or clarifications you might need.
>
> Thank you!

---

> > ### Comment · Reviewer_c19t · 2024-11-30
> >
> > Dear authors, thank you for these clarifications - I will however maintain my score.

---

### Author Response · Authors · 2024-11-26
**Brief Summary of Changes and Improvements**

We’d like to thank all reviewers for dedicating time to provide us with constructive and detailed feedback. We have made significant improvement on the quality of our paper and highlight some important improvements.

**Primary Changes:**
1. We include ablations to highlight TTRAG’s performance as a comprehensive system followed by individual experiments to quantify each component's contribution. We observe significant performance gains across different datasets and present detailed results in Table 1: HotpotQA (+17.29%), QASPER (+4.39%), and Natural Questions (+8.73%).

**Secondary Changes:**
1. Improved writing mechanics, more informative figures, now refer to Test-Time RAG as ‘TTRAG’
2. Reran ablations for Llama-models with the standardized prompt template (specific to Llama models) rather than normal prompting and updated results in the relevant tables.
3. Moved empirical examples for conditional embeddings, query rewriting, and detailed results for conditional compute to appendix to improve readability and accommodate space.

---

### Meta-Review · Area_Chair_WsWW · 2024-12-19

**Metareview:**

This paper introduces a novel framework for enhancing long-context understanding in language models through dynamic document representation, query rewriting, and task-adaptive processing. The approach shows improvements across several question-answering benchmarks with varying context lengths.

The research demonstrates significant strengths in addressing the challenge of handling long-context information, with comprehensive evaluation across multiple datasets and model architectures. The authors provide valuable analysis of efficiency gains compared to baseline approaches. However, the paper faces substantial challenges in both presentation and methodology. The writing lacks clarity, making key concepts and methodologies difficult to understand. There appears to be misalignment between the stated objectives regarding personalized AI and the actual focus on long-context benchmarks.

Technical concerns include insufficient justification for the processing approach, unclear integration between components, and questions about practical feasibility due to computational requirements. The query rewriting method also raises potential issues about accuracy that could affect the system's reliability. The paper omits several crucial elements, including comprehensive literature review on related techniques, detailed performance analysis, clear iteration criteria, and statistical validation of improvements. The evaluation methodology tests components in isolation rather than as an integrated system, leaving questions about overall effectiveness.

Given these limitations, the current submission does not meet conference standards despite addressing an important problem with some promising results. The combination of unclear presentation, incomplete technical details, and questions about practical implementation suggests the need for substantial revision. Future work should focus on improving clarity, providing stronger technical justification, conducting more thorough empirical analysis, and offering comprehensive comparisons with existing methods.

**Additional Comments On Reviewer Discussion:**

The paper received critical feedback from multiple reviewers who identified several significant concerns. The review process revealed issues with the literature review's comprehensiveness, terminology usage, and the appropriateness of selected benchmarks. Concerns were also raised about the quality of writing, inconsistency between the introduction and main content, and integration challenges among the system's components. Additionally, practical implementation aspects were questioned, particularly regarding computational efficiency and potential problems with the query rewriting mechanism.

In response, the authors made several modifications to address these concerns. They introduced ablation studies to demonstrate their system's effectiveness both as an integrated solution and through component-level analysis, reporting considerable performance improvements across multiple datasets. The authors also enhanced the writing quality, updated visual elements, standardized terminology, and reorganized the paper's structure by relocating detailed results to the appendix.

Despite these improvements, several fundamental issues remained unresolved. While the enhanced writing and additional experimental analysis represented positive developments, the paper still lacked a thorough comparison with existing approaches in the field. Questions persisted about the system's practical feasibility, computational requirements, and the alignment between stated goals and actual focus. The justification for certain processing approaches remained insufficient, and concerns about the query rewriting mechanism were not fully addressed. Although the revisions demonstrated progress, particularly through the addition of ablation studies validating system performance, the persistence of these core methodological and presentation issues ultimately supported maintaining the rejection recommendation. The authors were encouraged to address these fundamental concerns in a future submission.

---

### Decision · Program_Chairs · 2025-01-22

Reject